# Fair Synthetic Data Does not Necessarily Lead to Fair Models

## Abstract

The Wasserstein GAN (WGAN) is a well-established model allowing for the generation of high-quality synthetic data approximating a given real dataset. We study TabFairGAN, a known tabular variation of WGAN in which a custom penalty term is added to the generator's loss, forcing it to produce fair data. Here we measure the fairness of synthetic data using demographic parity, i.e., the gap in the proportions of positive outcome between different sensitive groups. We reproduce some results from the paper and highlight empirically the fact that although the synthetic data achieves low demographic parity, a classification model trained on said data and evaluated on real data may still output predictions that achieve high demographic parity – hence is unfair. In particular, we show empirically this gap holds for most parts spectrum of the fairness-accuracy tradeoff, besides the large-penalty case where the model mode collapses to the most frequent target outcome, and the low-penalty case where the data is not constrained to be fair.

## 1 Introduction

The problem of constructing fair machine learning models has been well studied, with a range of possible approaches, such as using different thresholds for different protected classes (Jang et al. [2022]), adding an optimal transport penalty to a neural network's loss function (Gordaliza et al. [2019]) and more. Among these approaches, generating synthetic data under fairness constraints is a prevalent one (see Xu et al. [2018, 2019], van Breugel et al. [2021], Jang et al. [2021]). A common way to evaluate the fairness of synthetic data is using demographic parity. Given a data set with sensitive binary attribute $S$ and binary target $Y$, the demographic parity of the data measures the dependence between the distributions of $S$ and $Y$. Similarly, the demographic parity of a classifier $f$ trained on said synthetic data measures the dependence between $S$ and predictions $f(X, S)$ where $X$ consists of the set of unsensitive variables. In this latter case, the evaluation of the classifier's fairness is done on the real data. Thus, training a classifier on the synthetic fair data and evaluating its demographic parity over a holdout of the real data gives us another viable way to measure the fairness of the data. In fact, if one's goal is to use synthetic fair data to train models which are fair in production, the latter metric is intuitively more relevant.

In this paper we focus on TabFairGAN (Rajabi and Garibay [2022]), a known variant of WGAN (Arjovsky et al. [2017]) in which a demographic parity penalty term is added to the generators loss function, forcing it to produce data with low dependence between $S$ and $Y$. We measure the fairness of the data generated by the aforesaid model in both ways, on two common benchmark datasets, and show that besides in trivial cases, the demographic parity of a classifier tends to be significantly higher than the demographic parity of the synthetic data it has been trained on for most of the fairness-accuracy spectrum. This implies that although the synthetic data generated by our model is fair, using this data for training may not be a good way to create models which produce similar output distributions when conditioned on different protected classes.

Submitted to 36th Conference on Neural Information Processing Systems (NeurIPS 2022). Do not distribute.

## 2 Background

### 2.1 Demographic Parity

Given a dataset $\mathcal{D} = \{X, s, y\}$ where the vector $X \in \mathbb{R}^n$ represents the unprotected attributes, $s \in \{0, 1\}$, is the protected attribute and $y \in \{0, 1\}$ is the target feature, the demographic parity of $\mathcal{D}$ is defined by:

$$DP_{\text{data}}(\mathcal{D}) = |P_{\mathcal{D}}(y = 1|s = 1) - P_{\mathcal{D}}(y = 1|s = 0)|. \tag{1}$$

where

$$P_{\mathcal{D}}(y = 1|s = 1) = \frac{\sum_{i=1}^{N} \mathbb{I}\{y_i = 1, s_i = 1\}}{\sum_{i=1}^{N} \mathbb{I}\{s_i = 1\}}, \quad P_{\mathcal{D}}(y = 1|s = 0) = \frac{\sum_{i=1}^{N} \mathbb{I}\{y_i = 1, s_i = 0\}}{\sum_{i=1}^{N} \mathbb{I}\{s_i = 0\}} \tag{2}$$

Given a classifier $f : \mathbb{R}^n \to \{0, 1\}$ trained on the dataset $\mathcal{D}_{\text{train}}$, and a test dataset $\mathcal{D}_{\text{test}}$, the demographic parity of $f$ is defined by:

$$DP_{\text{model}}(f, \mathcal{D}_{\text{test}}) = |P_{\mathcal{D}_{\text{test}}}(f(x, s) = 1|s = 1) - P_{\mathcal{D}_{\text{test}}}(f(x, s) = 1|s = 0)|. \tag{3}$$

where

$$P_{\mathcal{D}_{\text{test}}}(f(x, s) = 1|s = 1) = \frac{\sum_{i=1}^{N} \mathbb{I}\{f(x_i, s_i) = 1, s_i = 1\}}{\sum_{i=1}^{N} \mathbb{I}\{s_i = 1\}} \tag{4}$$

It is worth noting that the notion of demographic parity can be generalized to the case of multiple sensitive classes and multiple targets and that there exist many more notions of fairness (see Walby and Armstrong [2011]).

### 2.2 WGAN

The architecture of WGAN is based on the Kantorovich-Rubinstein duality (Edwards [2011]), a theorem in the field of optimal transport (Villani [2009]), and involves a generator $G$, and a critic $C$. The generator's role is to transform the distribution of a random noise $z \sim \mathcal{N}(0, 1)$ into a distribution $\mathbb{P}_g$ which approximates some real data distribution $\mathbb{P}_r$. The critic's role is to evaluate the quality of a given data point. The kantorovich Rubinstein duality suggests the critic should be approximately 1-Lipschitz and so we add a gradient penalty term to the discriminator's loss. Thus, the loss functions that the two models try to minimize are:

$$\mathcal{L}_{\text{C}} = -\mathbb{E}_{\tilde{x} \sim \mathbb{P}_g}[C(\tilde{x})] + \mathbb{E}_{x \sim \mathbb{P}_r}[C(x)] + \gamma \mathbb{E}_{\hat{x} \sim \mathbb{P}_{\hat{x}}}[(\|\nabla_{\hat{x}} C(\hat{x})\|_2 - 1)^2] \tag{5}$$

$$\mathcal{L}_{\text{G}} = -\mathbb{E}_{\tilde{x} \sim \mathbb{P}_g}[C(\tilde{x})] \tag{6}$$

where $\mathbb{P}_{\hat{x}}$ is a distribution of points, uniformly sampled along straight lines between pairs of points sampled from $\mathbb{P}_r$ and $\mathbb{P}_g$, $(\|\nabla_{\hat{x}} C(\hat{x})\|_2$ is the gradient norm of the critic's scores on $\hat{x}$, and $\gamma$ is the gradient penalty coefficient. The latter has been kept constant at the value of 10 throughout all of WGAN training trials, as recommended by the authors of the WGAN-GP paper (Gulrajani et al. [2017]). The main reason we use WGAN instead of the original GAN (Goodfellow et al. [2014]) is that it is less prone to mode collapse.

## 3 Methodology

### 3.1 Fair Data Generation

As previously mentioned, to produce fair synthetic data we use tabFairGAN, a common variant of WGAN which involves adding a fairness penalty term to the generator loss of a standard WGAN model. Thus the generator's loss becomes:

$$\mathcal{L}_{\text{G}} = -\mathbb{E}_{\tilde{x} \sim \mathbb{P}_g}[C(\tilde{x})] + \lambda DP_{\text{data}}^2(\mathcal{D}) \tag{7}$$

where $\mathcal{D} = \{x_i, s_i, y_i\}_{i=1}^N$ and $(x_i, s_i, y_i) \sim \mathbb{P}_g$. The second term is equal to the demographic parity of the generated data squared, and $\lambda$ is a hyper-parameter used to control the synthetic data's utility-fairness trade-off. As the generator is trained to minimize its loss, introducing the new penalty term forces the generator to produce data with lower demographic parity.

Following tabFairGAN, the generator's network consists has Gumbel-Softmax heads for categorical variables, however, we do not use variational Gaussian normalization for continuous variables.

## 3.2 Data

In order to test our hypothesis, we use the Census Adult data-set (Blake [1998]), and Compas Recidivism data-set (Angwin et al. [2016]), two common benchmarking data-sets in the fairness field – both considered in Rajabi and Garibay [2022]. The first is a classification problem of 48842 records, in which we try to predict whether a person earns over 50K dollars a year, using features like education, family status, etc. The sensitive attribute here is the gender of a person. The second is a classification problem of 16267 records where we try to predict whether ex-criminal offenders have a high or low chance to commit another crime, based on features like previous offense type, language, etc. Studies have shown that this dataset is biased against African Americans (Chouldechova [2017]). Therefore ethnicity is chosen to be the protected attribute for this study. We shall use an altered version of the original dataset which was used in the original TabFairGAN paper.

In both cases, the target and the protected attribute are binary. We pre-process the data using one-hot encoding on all categorical features and standard scaling on all numeric features.

## 3.3 Metrics

We concern ourselves with three properties of synthetically generated data, its utility, its population fairness, and the fairness of a model trained using it. Thus, the three metrics which we will concentrate on are the following: we will measure the data's utility using the accuracy of a logistic regression model trained on the synthetic data and assessed on a test set of real data. we will measure the data's population fairness using The demographic parity of the synthetic data itself, and finally, we will measure the fairness of a model trained on the data using the demographic parity of the output of a logistic regressions model trained on the synthetic data and evaluated on the test data.

## 4 Experiments

In the first experiment, we report accuracy and demographic parity scores on the Census dataset (see Figure 1).

As the fairness weight $\lambda$ grows, the accuracy of the classifier trained on the synthetic data decreases, starting with a value as high as that of a classifier trained on the real data, and ending with an accuracy close to that of a classifier which assigns only the outcome with the highest probability (this achieves an accuracy of 0.76), and has a perfect data demographic parity as a result. We also see that both the demographic parity of the synthetic data, and that of the logistic regression classifier trained on it, start at around 0.19, which is the demographic parity of the real data, and end at around zero.

However, more importantly, we observe a gap between data and model demographic parity. It is indeed required to pay a significantly higher price in accuracy to reach the same level of parity for synthetic data and model. We observe similar conclusions in the Compas experiment (see Figure 2). In this case, the gap is significantly higher – for an accuracy of $85\%$, the data demographic parity is at 0.02, while the model demographic parity is of 0.17. This suggests that care need to be taken when leveraging synthetic data achieving fairness constraints as this does not necessarily means a model trained on said data, and tested on a real test holdout will lead to fair predictions.

Also, the only places where the gap is minor is when the fairness weight is close to 0, in which case the data is not constrained to be fair, and the case where the fairness weight becomes very large, in which case the classification model simply mode-collapses to the most frequent target outcome, (and hence achieves perfect model demographic parity as can be observed in Figure 1).

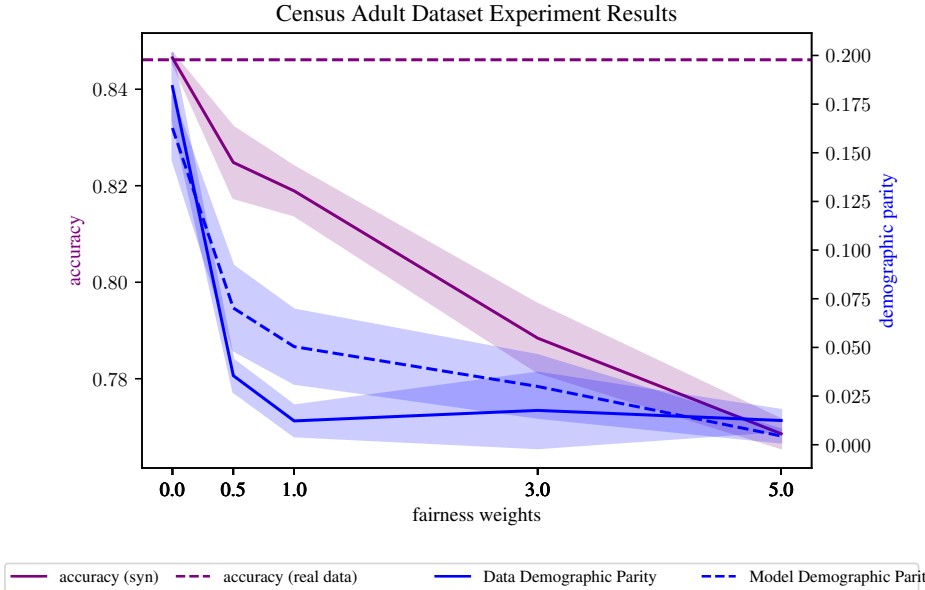

Figure 1: The gap between the demographic parity of synthetic data generated by TabFairGan and the demographic parity of a logistic regression classifier trained on said data and evaluated on real data. Here the results are evaluated on the Census Adult dataset. The results were evaluated over 10 seeds and are presented with a $95\%$ confidence interval.

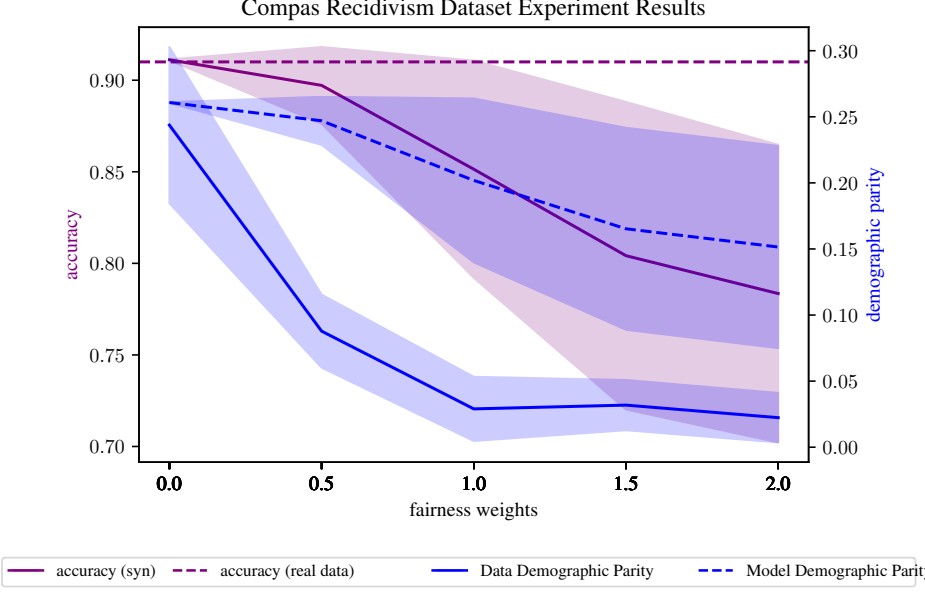

Figure 2: The gap between the demographic parity of synthetic data generated by TabFairGan and the demographic parity of a logistic regression classifier trained on said data and evaluated on real data. Here the results are evaluated on the Compas Recidivism dataset. The results were evaluated over 10 seeds and are presented with a $95\%$ confidence interval.

## 5 Discussion

Demographic parity is solely dictated by the correlation between the target variable $Y$ and the sensitive variable $S$. However, it does not account for proxy variables, and can be low despite the data

having other dependencies, which may bias the results. This implies that a synthetic data generation model which is trained with demographic parity constraints, can achieve these constraints by changing $P(X, s)$ rather then changing $P(y|X, s)$ (i.e. changing the population itself rather then the targets dependency on the population). This drift leads to a fairness gap, i.e., the training data may look artificially fair, but a model trained on it and deployed (tested) on a holdout dataset from the true population can still lead to unfair predictions.

## 6  Conclusion

In this paper, we reproduce and study a minor variation of TabFairGAN, a recently proposed approach to train GANs under demographic parity constraints. We observed that while this class of model allows to achieve low data demographic parity with negligible accuracy drop, achieving downstream model demographic parity is more challenging. This suggests that care should be taken when leveraging synthetic data with demographic parity constraints as models trained on said data may not be downstream-fair.

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
