# OpenReview forum: "Fair Synthetic Data Does not Necessarily Lead to Fair Models"
_NeurIPS.cc/2022/Workshop/SyntheticData4ML — Neurips 2022 SyntheticData4ML_

### Official Review · Reviewer_wA9W · 2022-10-11
**Interesting setup!**

**Rating:** 6
**Confidence:** 4

**Review:**

I thank the reviewers for submitting a paper such as this one. I believe this is important work and should receive more attention. However, I have two minor remarks:
- I would have enjoyed a more intuitive explanation/conclusion for the behaviour we observe in the experiments. In fact, it seems almost expected that DP would have the same performance as the data when lambda=0? Is this a bad thing?
- As the authors compare using only one fair synth data generator, I wonder whether the observed experiments are merely an artifact of that specific method, or whether there is a bigger point to be made here. Of course, I understand that this being a workshop, we cannot expect a full sweep of all the existing methods. However, one or two additional ones may actually result in an entirely different conclusion.

---

### Official Review · Reviewer_GqDU · 2022-10-18
**Needs additional references**

**Rating:** 7
**Confidence:** 2

**Review:**

The paper is clearly written and experimental setup is thoroughly explained. Explaining mode collapse in separate section might increase readability for the reader. Additional paper could have been referenced to set proper context for novelty of the paper (one example below).

https://ojs.aaai.org/index.php/AAAI/article/view/16965

The results could have been compared with other approaches on same datasets and evaluation setup.

---

### Official Review · Reviewer_vRkP · 2022-10-18
**Interesting paper  but without any insights**

**Rating:** 4
**Confidence:** 3

**Review:**

The paper studies the impact of synthetic, fair data on classifiers. The authors propose using a TabFairGAN to train GANs  under demographic parity assumption. While this helps generate data with low demographic parity, these gains are not transferred when learning models.  Experiments are performed to investigate demographic parity of downstream models.  However, no insights into this phenomenon or explanations have been offered. Perhaps one reason for this might be as follows:

While demographic parity might help ensure low correlation of a certain variable (say S) with the label, there could be information leakage via other certain variables that are correlated with S. These correlations, when apriori not controlled might lead to disappearing demographic parity at the model level.

Could the authors provide some intuition?

---

### Meta-Review · Area_Chair_Mxz3 · 2022-10-20

**Recommendation:** Accept

**Review:**

Though the reviewers agree that the paper could have contained more insight, the conclusion seems interesting enough and very relevant to the community. I recommend to accept it to the workshop.